# Attack tolerant fault diagnosis based on unknown input interval observer

1st Qidong Liu, 2nd Yue Long†, 3rd Tie-Shan Li

*School of Automation Engineering*
*University of Electronic Science and Technology of China*
Chengdu, China
longyue@uestc.edu.cn

*Abstract*—For cyber-physical systems that are adversely affected by actuator attacks, faults, and disturbances, this paper proposes a secure and robust fault estimator designed for real-time detection and estimation of potential fault signals, laying the groundwork for subsequent secure fault tolerance strategies. Specifically, the designed fault estimator integrates the features of unknown input observers and interval observers, enabling it to decouple disturbance signals and generate upper and lower bounds for the fault signal even in the presence of actuator attacks. Through mathematical derivation, linear solvable conditions have been provided that ensure the stability of the nominal error system, non-negativity of the error system matrix, and robustness of the system under attack. Finally, a set of simulation experiments demonstrates the effectiveness of this fault diagnosis method.

*Index Terms*—Resilient fault diagnosis, unknown input interval observer, actuator attack.

## I. INTRODUCTION

In recent years, with the advancement of 3C technology, the influence of the cyber domain has been continuously expanding, leading to an increase in the scale of networked systems and enhancements in system functionalities. Cyber-Physical System (CPS), representing the core of the next generation of technology, have emerged as a focal point of research. Unlike traditional control systems, CPS not only integrate computing units with physical entities closely but also emphasize the interconnectivity among devices, thereby enhancing system adaptability, scalability, and productivity. Significant achievements have been realized in various domains such as smart grids, smart healthcare, and intelligent transportation systems [1], [2]. However, the capabilities of CPS for intelligent perception, decision-making, and analysis rely on the deep integration of cyber and physical spaces, which not only increases system complexity but also introduces a plethora of risks. The safety and reliability of engineering systems are of paramount importance for CPS due to the interactive nature of systems that could potentially spread anomalies from a single node across the entire network, leading to severe consequences. Therefore, a deep understanding of potential threats to CPS and the design of corresponding anomaly detection and response strategies are indispensable components in the development of CPS.

This work is supported in part by the National Natural Science Foundation of China under Grants 62273072, 51939001.

One of the novel challenges that CPS face is cyber-attacks, which often manipulate the network layer to impact the real world [3], [4]. These attacks include, but are not limited to, denial of service, spoofing, and channel availability attacks. While encryption and decryption methods in IT technology are effective in preventing information tampering, they inevitably reduce the efficiency of network communications and fail to fully exploit the interactive information between physical devices and the network. Consequently, reliance solely on IT technology is insufficient to ensure the secure and stable operation of CPS in the face of cyber-attacks.

Current research on CPS under cyber-attacks focuses on several key areas. Firstly, system security analysis is emphasized [5], encompassing vulnerability assessments and the creation of perfect attack signals. Moreover, attention is given to secure state estimation [6], where the aim is to develop information filtering mechanisms that can swiftly and accurately calculate the most precise state estimates from contaminated sensor data. Furthermore, the focus shifts to resilient control strategies [7], which leverage incomplete or corrupted output or state information, utilizing feedback and compensation mechanisms to ensure the system's availability and maintain the integrity of its functionalities. Undertaking research and development in these critical areas of CPS is thus imperative, as it underpins the necessity to enhance their resilience and functionality in the face of cyber threats.

However, it is important to note that, as an integration of previously complex systems, CPS are also subject to adverse effects from traditional threats such as multi-source disturbances, system failures, or sensor faults [8], [9]. How to maintain the safety of the system in the presence of multiple threats like faults and attacks simultaneously is an issue that cannot be overlooked. Some research exists on this topic such as [10]–[13]. Specifically, [10] investigates anomaly detection mechanisms in CPS confronted with multiple attacks and faults, while [11] focuses on fault estimation problem under intermittent DoS attacks. Additionally, [12] studies secure fault-tolerant control problem when coping with sparse sensing attacks, and [13] explores fault-tolerant control strategies during multi-channel DoS attacks, illustrating the varied and crucial efforts to enhance CPS security. However, there is a relative scarcity of work addressing fault estimation in CPS under the influence of unknown forms of actuator attacks.

Inspired by the aforementioned work, this paper extends research on state estimation by exploring the fault estimation issue through unknown input interval observer (UIIO). The main contribution of this study is the introduction of an enhanced UIIO designed for precise fault signal estimation under simultaneous faults, cyber-attacks, and system disturbances. It delineates the upper and lower bounds for the fault signals while not requiring consideration of the specific forms of actuator attacks, providing a robust basis for advancing secure control methodologies.

The remainder of the article provides an analysis and presents linear solvable conditions that ensure the stability, robustness, and non-negative characteristics of the augmented error system's matrix. Moreover, a set of simulation results confirm the viability of the suggested approach.

## II. PROBLEM STATEMENT

### A. Fuzzy CPS

Consider the following discrete-time fuzzy system subject to actuator attack, disturbance and fault

$$x(k+1) = \sum_{h=1}^{\nu} \phi_h(y(k))[A_h x(k) + B_{h1} u^a(k)$$
$$+ B_d d(k) + B_{fh} f(k)] \quad (1)$$
$$y(k) = \sum_{h=1}^{\nu} \phi_h(y(k))[C x(k) + D_{fh} f(k)],$$

where $x(k) \in \mathbb{R}^n$ and $y(k) \in \mathbb{R}^m$ denotes vector of system state and output signal. $u^a(k)$ is the actuator input suffer from actuator attack and other additional signals include the decoupled disturbance and fault represented by $d(k) \in \mathbb{R}^d$ and $f(k) \in \mathbb{R}^f$, respectively. $A_h$, $B_{h1}$, $B_d$, $B_{fh}$, $C$ and $D_f$ are known constant matrices.

Consider that the output space is divided as [14], then (1) can be rewritten as

$$x(k+1) = \sum_{i \in O_2} \phi_i(y)[A_i x(k) + B_{i1} u^a(k)$$
$$+ B_d d(k) + B_{fi} f(k)]$$
$$y(k) = \sum_{i \in O_2} \phi_i(y)[C x(k) + D_{fi} f(k)], g \in O_1, y(k) \in S_g. \quad (2)$$

where $\phi_i(y) = \phi_i(y(k)) > 0$, $\sum_{i \in O_2(g)} \phi_i(y) = 1$ and $S_g$ is the separation and $g \in O_1$. $f(k)$ involves actuator fault and sensor fault with forward difference $\Delta f(k)$. Let $\bar{x}(k) = [x(k); f(k)]$, $u(k) = u^a(k) - a(k)$ with $a^-(k) \leq a(k) \leq a^+(k)$ then we have

$$\bar{x}(k+1) = \sum_{i \in O_2} \phi_i(y)[\bar{A}_i \bar{x}(k) + \bar{B}_{i1} u^a(k)$$
$$+ \bar{B}_d d(k) + \bar{B}_{d2i} d_2(k)], \quad (3)$$
$$y(k) = \sum_{i \in O_2} \phi_i(y)[\bar{C}_i \bar{x}(k)]$$

where $\bar{A}_i = \begin{bmatrix} A_i & B_{fi} \\ 0 & (1-\theta)I \end{bmatrix}$, $\bar{B}_{i1} = \begin{bmatrix} B_{i1} \\ 0 \end{bmatrix}$, $\bar{B}_d = \begin{bmatrix} B_d \\ 0 \end{bmatrix}$, $\bar{C}_i = \begin{bmatrix} C & D_{fi} \end{bmatrix}$, $\bar{B}_{d2i} = [0; \ I]$, $d_2(k) = \theta f(k) + \Delta f(k)$, $0 < |\theta| < 1$.

**Remark 1.** Given the unknown capabilities of attackers posing threats to CPS, this article makes no assumptions regarding the forms of C-A attacks considered, meaning that it does not confine its scope to previously identified types such as deception attacks, denial-of-service attacks, etc. On the other hand, from the perspective of system architecture, it is often possible to obtain extreme information about actuator mechanisms, such as the maximum angle of a rudder or the maximum power of an engine. Hence, it is logical to presume the availability of information on $a^+(k)$ and $a^-(k)$.

**Remark 2.** In order to equip the designed UIIO with the capability to estimate faults, this study integrates the scalar $\theta$ into its framework, leveraging the methodology outlined in [15].

### B. Unknown Input Interval Observer

The designed UIIO is capable, on one hand, of isolating disturbance signals, and on the other hand, of utilizing the bound information of the actuator attack signals to estimate fault signals, even in the presence of actuator attacks.

$$\begin{cases} \bar{z}^+(k+1) = \sum_{i \in O_2} \phi_i(y)[\bar{T}(\bar{A}_i \hat{\bar{x}}^+(k) + \bar{B}_{1i} u(k)) \\ \quad + \bar{K}_h^+(\hat{y}^+(k) - y(k)) + \bar{J}_h^+(\hat{\bar{x}}^+(k) - \hat{\bar{x}}^-(k)) \\ \quad + \check{B}_{2i} \check{d}_2(k) + \check{B}_{1i} \check{a}(k)] \\ \bar{z}^-(k+1) = \sum_{i \in O_2} \phi_i(y)[\bar{T}(\bar{A}_i \hat{\bar{x}}^-(k) + \bar{B}_{1i} u(k)) \\ \quad + \bar{K}_h^-(\hat{y}^-(k) - y(k)) - \bar{J}_h^-(\hat{\bar{x}}^+(k) - \hat{\bar{x}}^-(k)) \\ \quad + \hat{B}_{2i} \check{d}_2(k) + \hat{B}_{1i} \check{a}(k)] \end{cases} \quad (4)$$

$$\begin{cases} \hat{\bar{x}}^+(k) - \bar{H} y(k) = \bar{z}^+(k) \\ \hat{\bar{x}}^-(k) - \bar{H} y(k) = \bar{z}^-(k) \\ \hat{y}^+(k) = \sum_{i \in O_2} \phi_i(y)[\bar{C}_i^+ \hat{\bar{x}}^+(k) - \bar{C}_i^- \hat{\bar{x}}^-(k)] \\ \hat{y}^-(k) = \sum_{i \in O_2} \phi_i(y)[\bar{C}_i^+ \hat{\bar{x}}^-(k) - \bar{C}_i^- \hat{\bar{x}}^+(k)] \end{cases} \quad (5)$$

where $\check{B}_{1i} = [(\bar{T}\bar{B}_{1i})^+ - (\bar{T}\bar{B}_{1i})^-]$, $\check{B}_{2i} = [(\bar{T}\bar{B}_{2i})^+ - (\bar{T}\bar{B}_{2i})^-]$, $\hat{B}_{1i} = -\check{B}_{1i}$, $\hat{B}_{2i} = -\check{B}_{2i}$, $\check{a}(k) = [a^+(k); a^-(k)]$, $\check{d}_2(k) = [d_2^+(k); d_2^-(k)]$.

Define $e^+(k) = \hat{\bar{x}}^+(k) - \bar{x}(k)$, $e^-(k) = \bar{x}(k) - \hat{\bar{x}}^-(k)$. Upon computing the forward difference for $e^+(k)$ and $e^-(k)$, it follows that if conditions like

$$\begin{bmatrix} I & 0 \end{bmatrix} = \begin{bmatrix} \bar{H} & \bar{T} \end{bmatrix} \begin{bmatrix} \bar{C}_i & 0 \\ I & \bar{B}_d \end{bmatrix} \quad (6)$$

can be satisfied, then following equations are obtained

$$
\begin{cases}
e^+(k+1) = \sum_{i \in O_2} \phi_i(y)[(\bar{T}\bar{A}_i + \bar{K}_h^+ \bar{C}_i^+ + \bar{J}_h^+)e^+(k) \\
\qquad + (\bar{J}_h^+ + \bar{K}_h^+ \bar{C}_i^+)e^-(k) \\
\qquad + \check{B}_{1i}\check{a}(k) - \bar{T}\bar{B}_{1i}a(k) + \check{B}_{2i}\check{d}_2(k) - \bar{T}\bar{B}_{2i}d_2(k)], \\
e^-(k+1) = \sum_{i \in O_2} \phi_i(y)[(\bar{T}\bar{A}_i + \bar{K}_h^- \bar{C}_i^+ + \bar{J}_h^-)e^-(k) \\
\qquad + (\bar{J}_h^- + \bar{K}_h^- \bar{C}_i^-)e^+(k) \\
\qquad + \bar{T}\bar{B}_{1i}a(k) - \hat{B}_{1i}\check{a}(k) + \bar{T}\bar{B}_{2i}d_2(k) - \hat{B}_{2i}\check{d}_2(k)]
\end{cases}
$$
(7)

$$
\begin{cases}
r^+(k) = \sum_{i \in O_2} \phi_i(y)[\bar{C}_i^+ e^+(k) + \bar{C}_i^- e^-(k)] \\
r^-(k) = \sum_{i \in O_2} \phi_i(y)[\bar{C}_i^+ e^-(k) + \bar{C}_i^- e^+(k)]
\end{cases}
$$
(8)

**Lemma 1.** For system (2), the condition for the existence of UIIO (4)(5) that is with property (6) is given as
1) rank$(\bar{C}_i \bar{B}_d)$= rank$(\bar{B}_d)$,
2) $(\bar{C}_i, \bar{T}\bar{A}_i)$ is a detectable pair.
and $\bar{H}$ and $\bar{T}$ can be calculated by

$$
[\bar{T} \quad \bar{H}] = [I \quad 0]\mathfrak{B} + \mathfrak{A}(I - \mathfrak{B}\mathfrak{B}^\dagger)
$$
(9)

where $\mathfrak{A}$ is an arbitrary matrix and $\mathfrak{B} = \begin{bmatrix} I & \bar{B}_d \\ \bar{C}_i & 0 \end{bmatrix}$.

As stated in [16, Lemma 2, Lemma 3], condition

$$
\check{B}_{1i}\check{a}(k) - \bar{T}\bar{B}_{1i}a(k) \geq 0, \bar{T}\bar{B}_{1i}a(k) - \hat{B}_{1i}\check{a}(k) \geq 0,
$$
$$
\check{B}_{2i}\check{d}_2(k) - \bar{T}\bar{B}_{2i}d_2(k) \geq 0, \bar{T}\bar{B}_{2i}d_2(k) - \hat{B}_{2i}\check{d}_2(k) \geq 0.
$$
(10)

can be satisfied.

Let $\tilde{e} = \begin{bmatrix} e^+ \\ e^- \end{bmatrix}$, $\tilde{d} = \begin{bmatrix} \tilde{d}_1 \\ \tilde{d}_2 \end{bmatrix}$, $\tilde{r} = \begin{bmatrix} \tilde{r}_1 \\ \tilde{r}_e \end{bmatrix}$, $\tilde{r}_1 = \begin{bmatrix} r^+ \\ r^- \end{bmatrix}$, $\tilde{r}_e = \begin{bmatrix} e_f^+ \\ e_f^- \end{bmatrix}$, $\tilde{d}_1 = \begin{bmatrix} a - a^- \\ a^+ - a \end{bmatrix}$, $\tilde{d}_2 = \begin{bmatrix} d_2 - d_2^- \\ d_2^+ - d_2 \end{bmatrix}$, $C_f = [0, 0, I]$, $e_f^+ = C_f e^+$, $e_f^- = C_f e^-$, then we have:

$$
\begin{cases}
\tilde{e}(k+1) = \sum_{i \in O_2} \phi_i(y)[\tilde{A}_i \tilde{e}(k) + \tilde{B}_{di}\tilde{d}(k)] \\
\tilde{r}(k) = \sum_{i \in O_2} \phi_i(y)[\tilde{C}_i \tilde{e}(k)]
\end{cases}
$$
(11)

where $\tilde{A}_i = \begin{bmatrix} \bar{T}\bar{A}_i + \bar{K}_h^+ \bar{C}_i^+ + \bar{J}_h^+ & \bar{J}_h^+ + \bar{K}_h^+ \bar{C}_i^- \\ \bar{J}_h^- + \bar{K}_h^- \bar{C}_i^- & \bar{T}\bar{A}_i + \bar{K}_h^- \bar{C}_i^+ + \bar{J}_h^- \end{bmatrix}$,
$\tilde{B}_{di}^T = \begin{bmatrix} \tilde{B}_{1i}^T \\ \tilde{B}_{d2i}^T \end{bmatrix}$ $\tilde{B}_{1i} = \begin{bmatrix} (\bar{T}\bar{B}_{1i})^+ & (\bar{T}\bar{B}_{1i})^- \\ (\bar{T}\bar{B}_{1i})^- & (\bar{T}\bar{B}_{1i})^+ \end{bmatrix}$, $\tilde{B}_{2i} = \begin{bmatrix} (\bar{T}\bar{B}_{2i})^+ & (\bar{T}\bar{B}_{2i})^- \\ (\bar{T}\bar{B}_{2i})^- & (\bar{T}\bar{B}_{2i})^+ \end{bmatrix}$, $\tilde{C}_i^T = \begin{bmatrix} \bar{C}_i^{+T} & \bar{C}_i^{-T} & C_f^T & 0 \\ \bar{C}_i^{-T} & \bar{C}_i^{+T} & 0 & C_f^T \end{bmatrix}$.

The main idea of fault estimation is to design the UIIO such that
1) Nominal error system is Schur stable and and possesses a Metzler system matrix.
2) System (11) is with finite frequency (FF) $H_\infty$ performance index $\sigma$ for disturbance $\tilde{d}(k) \in l_2[0, \infty)$.

**Remark 3.** As stated by [16, Lemma 2], submatrices $\bar{T}\bar{A}_i + \bar{K}_h^+ \bar{C}_i^+ + \bar{J}_h^+ \geq 0$ , $\bar{J}_h^+ + \bar{K}_h^+ \bar{C}_i^- \geq 0$ and $\bar{J}_h^- \bar{K}_h^- + \bar{C}_i^- \geq 0$,

$\bar{T}\bar{A}_i + \bar{K}_h^- \bar{C}_i^+ + \bar{J}_h^- \geq 0$, indicates that observers (7) and (8) function as an UIIO for system (2).

**Remark 4.** The purpose of designing the UIIO in this paper is to enable the estimation of fault signals, thereby facilitating compensatory control based on these estimations. Therefore, the additional term incorporated into the second condition $\tilde{r}(k)$ represents the error between the actual fault signal $f(k)$ and its estimated values $C_f \hat{x}^+(k), C_f \hat{x}^-(k)$.

## III. MAIN RESULTS

This section will provide linear solvable conditions that can ensure three aspects of performance: system stability, robustness, and the non-negativity of each element in the system matrix.

### A. Stability Analysis

This subsection specifies the sufficient conditions to achieve stability for system (11) via Linear Matrix Inequalities (LMI).
**Theorem 1.** System (11) is schur stable for all $h, j \in O_1$ $i \in O_2(h)$ if there exist positive definite matrices $P_{sh} = P_{sh}^T \succ 0$, $M_{sh}$, and matrices $\tilde{\bar{K}}_h^+$, $\tilde{\bar{K}}_h^-$, $\tilde{\bar{J}}_h^+$, $\tilde{\bar{J}}_h^-$ such that

$$
\begin{bmatrix} \begin{bmatrix} P_{sj1} - He\{M_{h1}\} & * & \\ P_{sj2} & P_{sj3} - He\{M_{h2}\} & \\ N_{i11} & \tilde{J}_h^{-T} + \bar{C}_i^{-T}\tilde{K}_h^{-T} & \\ \tilde{J}_h^{+T} + \bar{C}_i^{-T}\tilde{K}_h^{+T} & N_{i12} \end{bmatrix} & * \\ & -\begin{bmatrix} P_{sh1} & * \\ P_{sh2} & P_{sh3} \end{bmatrix} \end{bmatrix}
$$
$$
\prec 0
$$
(12)

where
$$
N_{i11}^T - \tilde{\bar{J}}_h^+ = M_{h1}\bar{T}\bar{A}_i + \tilde{\bar{K}}_h^+ \bar{C}_i^+,
$$
$$
N_{i12}^T - \tilde{\bar{J}}_h^- = M_{h2}\bar{T}\bar{A}_i + \tilde{\bar{K}}_h^- \bar{C}_i^+.
$$
(13)

**Proof:** Under the zero-initial condition,considering the augment system (11) when $\tilde{d}(k) = 0$, and selecting the following Lyapunov functional $V_s(k) = \tilde{e}(k)^T P_{sh}\tilde{e}(k)$ with

$$
\Delta V_s(k) = V_s(k+1) - V_s(k)
$$
$$
= \begin{bmatrix} \sum_{i \in O_2}(\tilde{A}_i \tilde{e}(k)) \\ \tilde{e}(k) \end{bmatrix}^T \begin{bmatrix} P_{sj} & 0 \\ 0 & -P_{sh} \end{bmatrix} \begin{bmatrix} \sum_{i \in O_2}(\tilde{A}_i \tilde{e}(k)) \\ \tilde{e}(k) \end{bmatrix}.
$$
(14)

Thus $\Delta V_s(k) \prec 0$ can be ensured when

$$
\begin{bmatrix} \sum_{i \in O_2}(\tilde{A}_i) \\ I \end{bmatrix}^T \begin{bmatrix} P_{sj} & 0 \\ 0 & -P_{sh} \end{bmatrix} \begin{bmatrix} \sum_{i \in O_2}(\tilde{A}_i) \\ I \end{bmatrix} \prec 0.
$$

Based on the Projection Lemma [16], $\Delta V_s(k) \prec 0$ is true if and only if a matrix $M_{sh}$ exists such that

$$
\begin{bmatrix} P_{sj} & 0 \\ 0 & -P_{sh} \end{bmatrix} + He \left\{ \begin{bmatrix} -I \\ \sum_{i \in O_2}(\tilde{A}_i) \end{bmatrix} \begin{bmatrix} M_{sh} \\ 0 \end{bmatrix}^T \right\} \prec 0.
$$
(15)

Let $M_{sh} = blockdiag\{M_{h1}, M_{h2}\}$, $\begin{bmatrix} \bar{K}_h^{+T} & \bar{J}_h^{-T} \\ \bar{J}_h^{+T} & \bar{K}_h^{-T} \end{bmatrix} M_{sh}^T = \begin{bmatrix} \tilde{\bar{K}}_h^{+T} & \tilde{\bar{J}}_h^{-T} \\ \tilde{\bar{J}}_h^{+T} & \tilde{\bar{K}}_h^{-T} \end{bmatrix}$, then (12) can be obtained. Conversely, demonstrating the system's stability is straightforward when (12) holds, thereby concluding the proof.

## B. Robustness of Error System

This subsection explores the FF $H_\infty$ performance of system (11). The premise is that the additive disturbance/attack signal operates within a low-frequency domain, as delineated in prior studies. Consequently, a theorem is presented herein.

**Theorem. 2** Given the matrices $\gamma_1$, $\gamma_2$, $\gamma_3$, $\gamma_4$, the defined interval augmentation error system (11) is with finite frequency $H_\infty$ norm $\sigma$ for every $h, j \in O_1$ and $i \in O_2(h)$, provided that there exist matrices $P_{rh} \succ 0$, $P_{rj} \succ 0$, $Q_{rh} \succ 0$, $M_{rh} \succ 0$, $\tilde{K}_h^+$, $\tilde{K}_h^-$, $\tilde{J}_h^+$, $\tilde{J}_h^-$ meeting the condition that

$$\begin{bmatrix} -P_{rj} & * & * \\ Q_{rh} - \Gamma M_{rh} & R_{rh} & * \\ 0 & (\tilde{B}_{di})^T M_{rh}^T \Gamma^T & -\sigma^2 I \end{bmatrix} \prec 0 \quad (16)$$

where $R_{rh} = P_{rh} - 2\cos(\iota_d)Q_{rh} + \tilde{C}_i^T \tilde{C}_i + N_{i2}$, $P_{rj} = \begin{bmatrix} P_{rj1} & * \\ P_{rj2} & P_{rj3} \end{bmatrix}$, $M_{rh} = M_{sh}$, $N_{i2} = \begin{bmatrix} N_{i21} & * \\ N_{i22} & N_{i23} \end{bmatrix}$,

$$N_{i21} = He\{N_{i11}\gamma_1^T + (\tilde{\tilde{J}}_h^{-T} + \bar{C}_i^{-T}\tilde{K}_h^{-T})\gamma_2^T\},$$
$$N_{i22} = (\tilde{\tilde{J}}_h^{+T} + \bar{C}_i^{-T}\tilde{K}_h^{+T})\gamma_1^T + N_{i12}\gamma_2^T$$
$$+ \gamma_3 N_{i11}^T + \gamma_4(\tilde{\tilde{J}}_h^{-T} + \bar{C}_i^{-T}\tilde{K}_h^{-T})^T,$$
$$N_{i23} = He\{N_{i12}\gamma_4^T + (\tilde{\tilde{J}}_h^{+T} + \bar{C}_i^{-T}\tilde{K}_h^{+T})\gamma_3^T\}.$$

**Proof:** As indicated by [16, Theorem 1], the condition below ensures that the error system (11) achieves an LF $H_\infty$ index $\sigma$.

$$\begin{bmatrix} S_1 \\ I_1 \end{bmatrix}^T \Omega_L \begin{bmatrix} S_1 \\ I_1 \end{bmatrix} + \begin{bmatrix} S_2 \\ I_2 \end{bmatrix}^T \begin{bmatrix} I & 0 \\ 0 & -\sigma^2 I \end{bmatrix} \begin{bmatrix} S_2 \\ I_2 \end{bmatrix} \prec 0 \quad (17)$$

where $S_1 = \left[ \sum_{i \in O_2}(\tilde{A}_i) \quad \sum_{i \in O_2}(\tilde{B}_{di}) \right]$, $S_2 = \left[ \sum_{i \in O_2}(\tilde{C}_i) \quad 0 \right]$, $I_1 = [I \; 0]$, $I_2 = [0 \; I]$.

Rewriting (17) we have

$$\begin{bmatrix} S_1 \\ I_1 \\ I_2 \end{bmatrix}^T \begin{bmatrix} -P_{rj} & * & * \\ Q_{rh} & P_{rh} - 2\cos(\iota_d)Q_{rh} + \tilde{C}_i^T\tilde{C}_i & * \\ 0 & 0 & -\sigma^2 I \end{bmatrix} \begin{bmatrix} S_1 \\ I_1 \\ I_2 \end{bmatrix} \prec 0 \quad (18)$$

According to the Projection Lemma, 18 holds iff $M_{rh}$ exists that fulfills condition

$$\sum_{i \in O_2} \left[ \begin{bmatrix} -P_{rj} & * & * \\ Q_{rh} & P_{rh} - 2\cos(\iota_d)Q_{rh} + \tilde{C}_i^T\tilde{C}_i & * \\ 0 & 0 & -\sigma^2 I \end{bmatrix} \right.$$
$$\left. + He\left\{ \begin{bmatrix} -I \\ \tilde{A}_i^T \\ \tilde{B}_{di}^T \end{bmatrix} M_{rh}^T [0 \quad \Gamma^T \quad 0] \right\} \right] \prec 0 \quad (19)$$

where $\Gamma^T = \begin{bmatrix} \gamma_1^T & \gamma_3^T \\ \gamma_2^T & \gamma_4^T \end{bmatrix}$. Selecting $M_{rh} = \begin{bmatrix} M_{h1} & * \\ 0 & M_{h2} \end{bmatrix}$ then (16) can be obtained, which completes the proof.

## C. Conditions to Ensure $\tilde{A}_i \geq 0$

It is crucial to verify that $\tilde{A}_i$ is a Metzler matrix. This subsection outlines how to ensure $\tilde{A}_i \geq 0$.

**Theorem 3.** $\tilde{A}_i \geq 0$ can be achieved if there are matrices $\bar{k}_{hp}$, $\underline{k}_{hp}$, $\mathfrak{j}_{hpq}$, $\bar{\mathfrak{j}}_{hpq}$, $M_{h1} = diag\{m_{h11}, \ldots, m_{h1\tilde{n}}\}$, $M_{h2} = diag\{m_{h21}, \ldots, m_{h2\tilde{n}}\}$ such that

$$m_{h1p}g_{ipq} + \bar{k}_{hp}\bar{c}_{iq} + \bar{\mathfrak{j}}_{hpq} \geq 0,$$
$$m_{h2p}g_{ipq} + \underline{k}_{hp}\bar{c}_{iq} + \mathfrak{j}_{hpq} \geq 0,$$
$$\bar{\mathfrak{j}}_{hpq} + \bar{k}_{hp}\underline{c}_{iq} \geq 0, \mathfrak{j}_{hpq} + \underline{k}_{hp}\underline{c}_{iq} \geq 0, \quad (20)$$
$$h \in O_1, i \in O_2(h), p, q = 1, 2, \ldots, \tilde{n}, \tilde{n} = n + f.$$

**Proof:** Let $(\bar{T}\bar{A}_i) = (g_{ipq})_{\tilde{n}\tilde{n}}$, $\bar{C}_i^+ = [\bar{c}_{i1}, \ldots, \bar{c}_{i\tilde{n}}]$, $\bar{C}_i^- = [\underline{c}_{i1}, \ldots, \underline{c}_{i\tilde{n}}]$, $\bar{K}_h^+ = [\bar{k}_{1h1}; \ldots; \bar{k}_{1h\tilde{n}}]$, $\bar{K}_h^- = [\underline{k}_{1h1}, \ldots, \underline{k}_{1h\tilde{n}}]$, $\bar{J}_h^- = (J_{hpq}^-)_{\tilde{n}\tilde{n}}$, $\bar{J}_h^+ = (J_{hpq}^+)_{\tilde{n}\tilde{n}}$, recalling that $\bar{T}\bar{A}_i + \bar{K}_h^+\bar{C}_i^+ + \bar{J}_h^+ \geq 0$, $\bar{J}_h^+ + \bar{K}_h^+\bar{C}_i^- \geq 0$, $\bar{J}_h^- + \bar{K}_h^-\bar{C}_i^- \geq 0$, $\bar{T}\bar{A}_i + \bar{K}_h^-\bar{C}_i^+ + \bar{J}_h^- \geq 0$, and these inequations can be rewritten as

$$\bar{T}\bar{A}_i + \bar{K}_h^+\bar{C}_i^+ + \bar{J}_h^+ = (g_{ipq} + l_{hipq}^+ + J_{hpq}^+)_{\tilde{n}\tilde{n}} \geq 0,$$
$$\bar{T}\bar{A}_i + \bar{K}_h^-\bar{C}_i^+ + \bar{J}_h^- = (g_{ipq} + l_{hipq}^- + J_{hpq}^-)_{\tilde{n}\tilde{n}} \geq 0,$$
$$\bar{J}_h^+ + \bar{K}_h^+\bar{C}_i^- = (J_{hpq}^+ + \bar{k}_{1hp}\underline{c}_{iq})_{\tilde{n}\tilde{n}} \geq 0, \quad (21)$$
$$\bar{J}_h^- + \bar{K}_h^-\bar{C}_i^- = (J_{hpq}^- + \underline{k}_{1hp}\underline{c}_{iq})_{\tilde{n}\tilde{n}} \geq 0,$$

where $l_{hipq}^+ = \bar{k}_{1hp}\bar{c}_{iq}$, $l_{hipq}^- = \underline{k}_{1hp}\bar{c}_{iq}$. Then $\tilde{A}_i \geq 0$ can be transformed into

$$g_{ipq} + l_{hipq}^+ + J_{hpq}^+ \geq 0, g_{ipq} + l_{hipq}^- + J_{hpq}^- \geq 0, \quad (22)$$
$$J_{hpq}^+ + \bar{k}_{1hp}\underline{c}_{iq} \geq 0, J_{hpq}^- + \underline{k}_{1hp}\underline{c}_{iq} \geq 0.$$

Due to the presence of coupled terms in Theorem 1 and Theorem 2, and the introduction of new variables $M_{rh}$, it is not straightforward to directly solve for $K_h^+$, $K_h^-$, $J_h^+$, and $J_h^-$. Consequently, adjustments as follows are necessary. Denoting

$$\tilde{K}_h^+ = M_{h1}\bar{K}_h^+ = [\bar{k}_{h1}; \ldots; \bar{k}_{h\tilde{n}}],$$
$$\tilde{K}_h^- = M_{h2}\bar{K}_h^- = [\underline{k}_{h1}; \ldots; \underline{k}_{h\tilde{n}}],$$
$$\tilde{\tilde{J}}_h^+ = M_{h1}\bar{J}_h^+ = [\bar{\mathfrak{j}}_{hpq}]_{\tilde{n}*\tilde{n}}, \tilde{\tilde{J}}_h^- = M_{h2}\bar{J}_h^- = [\mathfrak{j}_{hpq}]_{\tilde{n}*\tilde{n}},$$

multiplying the first and the third inequations of (21) with $M_{h1}$ and multiplying the second and the forth inequations of $M_{h2}$ from the left respectively, thereby transforming (22) into (20), which concludes the proof.

By addressing the optimization problem outlined as

$$\min \quad \sigma$$
$$s.t. \quad (12), (16), (20) \quad (23)$$

then gain matrices can subsequently be derived from

$$\begin{bmatrix} \bar{K}_h^{+T} & \bar{J}_h^{-T} \\ \bar{J}_h^{+T} & \bar{K}_h^{-T} \end{bmatrix} = \begin{bmatrix} \tilde{K}_h^{+T} & \tilde{\tilde{J}}_h^{-T} \\ \tilde{\tilde{J}}_h^{+T} & \tilde{K}_h^{-T} \end{bmatrix} M_{sh}^{T-1}. \quad (24)$$

## IV. EXAMPLE

This section validates the fault estimator derived from Theorems 1, 2, and 3, focusing on its ability to estimate two types of fault signals and its capability to track the system output. To demonstrate the efficacy of the fault estimation approach designed based on unknown input interval observer, the same fuzzy system model as referenced in [16] is utilized. While the system matrix and measurement matrix remain consistent,

modifications are exclusively applied to the gain matrices for the input, the disturbance and fault signals as detailed below:

$$B_d = \begin{bmatrix} -1 \\ 0.2 \end{bmatrix}, B_{11} = B_{21} = \cdots = B_1 = \begin{bmatrix} 1 \\ 0 \end{bmatrix},$$

$$B_{f1} = B_{f2} = \cdots = B_f = \begin{bmatrix} 0.12 \\ -0.18 \end{bmatrix},$$

$$D_{f1} = D_{f2} = \cdots = D_f = \begin{bmatrix} 0.29 \end{bmatrix}.$$

Firstly, the form of the disturbance signal is set as $d(k) = 0.5\sin(10k)$, the true control command is $u(k) = 0.1x(k)$, and the C-A attack is $a(k) = 0.3\cos(0.5k) + 0.1\sin(15k)$. Secondly, assuming two types of fault signals $f(k)$, the upper and lower bounds of the attack signal, $a^+(k)$ and $a^-(k)$, and the upper and lower bounds of the fault difference information, $d_2^+(k)$ and $d_2^-(k)$, are specified as follows:

$$f_1(k) = \begin{cases} 0.02(k-25), & k \in [25, 50], \\ 0.5 - 0.02(k-50) & k \in [51, 75], \\ 0 & \text{others}, \end{cases}$$

$$f_2(k) = \begin{cases} 0.5 sin(0.1\pi(k-25))e^{0.01(25-k)} & k \in [25, 75], \\ 0 & \text{others}, \end{cases}$$

$$a^+(k) = 0.7, a^-(k) = -0.7,$$

$$d_2^+(k) = 0.5(2d_2(k) + |\sin(k)|e^{(-0.05k)}),$$

$$d_2^-(k) = 0.5(2d_2(k) - |\sin(k)|e^{(-0.05k)}).$$

Next, the effects of fault estimation and output tracking under two different scenarios (corresponding to two distinct types of faults) will be presented.

**Scenario 1.** When the system is adversely affected by faults, disturbances, and attack signals simultaneously, the performance of the fault estimator designed in this paper is illustrated in Figure 1. It can be observed that due to significant difference in the prior information of the fault, the fault estimation error is considerable within the time interval $k \in [0, 20]$ and thereafter, it gradually converges to a smaller range. Similarly, the tracking values of the system output can also maintain satisfactory performance as shown in Figure 2.

**Scenario 2.** As shown in Figures 3 and 4, despite the system state is affected by fault signals $f_2$, yet the designed UIIO is still capable of stabilizing the error interval on both sides of 0, ensuring the accuracy of the observation interval. Similarly, due to the lack of precision in the prior information of the fault difference, the initial error is significant but gradually converges to a smaller interval.

It is important to note that in both sets of simulation results, the estimation error curves are very similar, due to the error system not being affected by system disturbances. In summary, when faced with two types of system fault signals, even though the state of the system is affected, the estimator designed based on the UIIO can still successfully estimate the fault signals within a certain period.

## V. CONCLUSION

To enhance the reliability and safety of systems, this paper investigates the fault estimation issue for fuzzy CPSs under

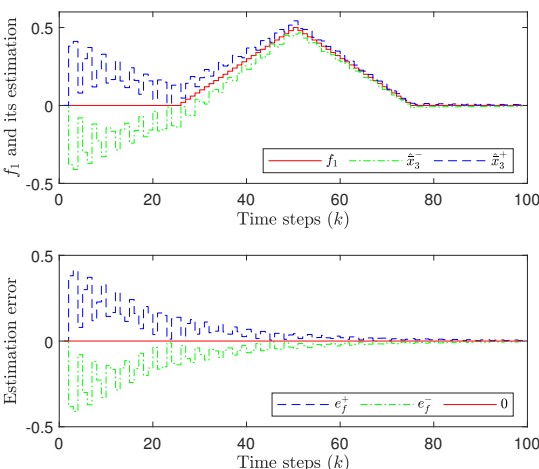

Fig. 1. Fault estimation performance and estimation error for $f_1(k)$

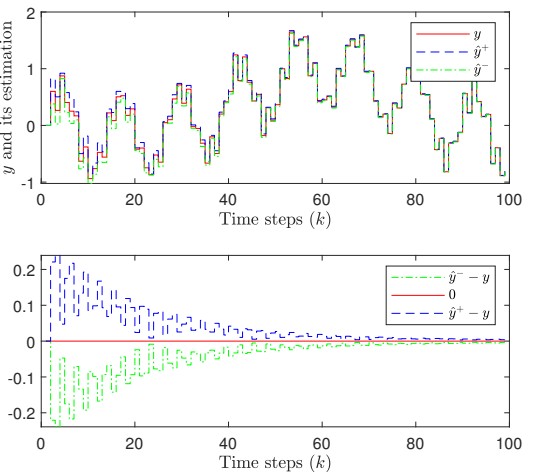

Fig. 2. Tracking performance and error of $y(k)$ under the influence of $f_1(k)$

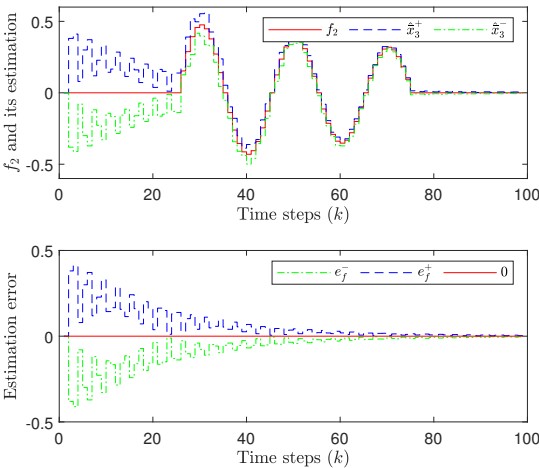

Fig. 3. Fault estimation performance and estimation error for $f_2(k)$

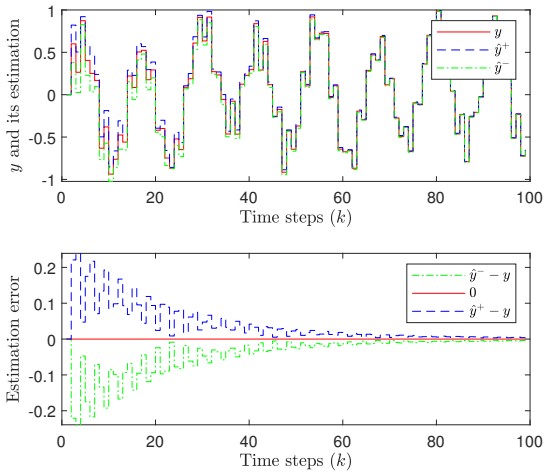

Fig. 4. Tracking performance and error of $y(k)$ under the influence of $f_2(k)$

the simultaneous influence of disturbances, faults, and actuator attacks, leading to the design of a fault estimator based on the UIIO. Specifically, this estimator is capable of decoupling disturbance signals, tolerating actuator attacks, and generating bounds for the fault signal interval. Through analysis, linear solvable conditions for deriving the estimator gains have been presented. Finally, the effectiveness of the designed estimator was validated using two different fault signals.

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
