# OpenReview forum: "Attack tolerant fault diagnosis based on unknown input interval observer"
_IEEE.org/ICIST/2024/Conference — IEEE ICIST 2024 Conference Submission_

### Official Review · Reviewer_4j8F · 2024-08-21
**This article is quite fascinating and of high quality.**

**Rating:** 7
**Confidence:** 3

**Review:**

The paper titled "Attack tolerant fault diagnosis based on unknown input interval observer" proposes a secure and robust fault estimator designed for real-time detection and estimation of potential fault signals, laying the groundwork for subsequent secure fault tolerance strategies. The designed fault estimator integrates the characteristics of an unknown input observer and an interval observer, enabling it to decouple disturbance signals and achieve effective disturbance rejection. My specific feedback is as follows: 1) How do the observers in the article achieve fault diagnosis. 2) Some formatting issues need to be addressed.

---

### Official Review · Reviewer_X97Z · 2024-08-21
**accept**

**Rating:** 7
**Confidence:** 3

**Review:**

Comment:
This paper investigated the fault estimation issue for fuzzy CPSs under the simultaneous influence of disturbances, faults, and actuator attacks. The theory is correct and can be accepted after responding the following comments.
(1)	In the introduction, it is not enough to state the current work. It should be expended and reconstructed.
(2)	There are many typos and grammar errors. The authors should have a native English speaker or software packages to perform the editing check.
(3)	The conclusion of the article suggests using the present perfect tense for description.

---

### Official Review · Reviewer_azJD · 2024-08-28
**Attack tolerant fault diagnosis based on unknown input interval observer**

**Rating:** 7
**Confidence:** 2

**Review:**

This paper investigates the fault estimation issue for fuzzy CPSs under the simultaneous influence of disturbances, faults, and actuator attacks, leading to the design of a fault estimator based on the UIIO.
a The abstract should mainly include elements such as research purpose, methods and final results, and reviewers suggest optimizing the content of the abstract.
b What are the significant differences between this study and previous studies? The author needs more explicit emphasis.
c There are some grammatical mistakes and typos. Please examine the full text further and revise them.

---

### Decision · Program_Chairs · 2024-09-06

Accept (Oral)